

# Metabolomic analysis of bioactive compounds in dill (*Anethum graveolens* L.) extracts

Jirattiporn Thanuma[1], Jutarop Phetcharaburanin[2,3], Hasaya Dokduang[4], Watcharin Loilome[2,5], Poramate Klanrit[2,5], Arporn Wangwiwatsin[2,5] and Nisana Namwat[2,5]

[1] Department of Biochemistry, Faculty of Medicine, Khon Kaen University, Khon Kaen City, Khon Kaen, Thailand

[2] Department of Systems Biosciences and Computational Medicine, Faculty of Medicine, Khon Kaen University, Khon Kaen City, Khon Kaen, Thailand

[3] KKU National Phenome Institute, Khon Kaen University, Khon Kaen City, Khon Kaen, Thailand

[4] Faculty of Medicine, Mahasarakham University, Mahasarakham, Mahasarakham, Thailand

[5] Cholangiocarcinoma Research Institute, Khon Kaen University, Khon Kaen City, Khon Kaen, Thailand

Corresponding author
Nisana Namwat, nisana@kku.ac.th

## ABSTRACT

**Background.** Dill (*Anethum graveolens* L.) is a functional herb known for its dietary, medicinal, and health-promoting agents, as it enriches antioxidants that help to protect cells from oxidative stress and may reduce the risk of several chronic diseases. The daily consumption of active components from dill, achieved through methods such as tea preparation and incorporation into soups and salads, relies on aqueous extraction techniques. The aim of this work was to investigate the metabolic profile of dill leaves extracted with water in various conditions. We also assessed the antioxidant and longevity properties of dill leaf extracts. The availability of aqueous-extracted chemicals from dill promotes therapeutic usage and nutritional supplements, showing its potential as a culinary and medicinal herb.

**Materials and Methods.** The metabolic profiles of dill leaves extracted with water at 27 °C and 90 °C were analyzed using untargeted proton nuclear magnetic resonance spectroscopy ($^1$H NMR). The one-dimensional (1D) followed by two-dimensional (2D) *J*-resolved $^1$H NMR were employed. The antioxidant activities of dill leaf extracts were measured using four methods: total phenolic content (TPC), ferric-reducing antioxidant power (FRAP), 2,2'-azino-bis(3-ethylbenzothiazoline-6-sulfonic acid) (ABTS), and oxygen radical absorbance capacity (ORAC) assays. The correlation between antioxidant properties and metabolites was analyzed using heatmap analysis. Reactive oxygen species (ROS) activity assays were conducted. The longevity effect of dill leaf extracts on human dermal fibroblasts (HDFs) was also examined using western blotting.

**Results.** The $^1$H NMR analysis identified 41 metabolites in dill leaf extracts, including 39 primary and two secondary metabolites. The antioxidant tests showed that an aqueous extraction of dill leaves with hot water at 90 °C resulted in much stronger antioxidant activities compared to using water at 27 °C. Antioxidant activities were positively correlated with the levels of five metabolites: pyridoxal, indole-3-lactate, adenine, inosine, and folate. β-Hydroxybutyrate, cellobiose, and α-glucose were the three metabolites that showed negative relationships with antioxidant activities. We also demonstrated that dill leaf extracts significantly reduced cytosolic oxidation and

altered longevity markers in human dermal fibroblasts (HDFs) by increasing FOXO3, SIRT6, and AMPK while diminishing levels of mTOR and Akt proteins.

**Conclusion**. Results indicated that dill leaf extracts contain antioxidant and anti-aging potential, especially in aqueous extracts at 90 °C. Key metabolites contributing to these effects might facilitate oxidative protection and cellular longevity in fibroblasts, supporting the value of dill as a dietary and medicinal herb.

## INTRODUCTION

Herbal plants have been used for medical and dietary purposes since ancient times due to their effectiveness, low toxicity, and safety for users compared to artificial agents (*Modak et al., 2007*). Herbal plants contain various types of phytochemicals with diverse biological functions. Consequently, many researchers worldwide make use of these medicinal benefits to achieve better health outcomes (*Said et al., 2002*). Globally, more than 250,000 plant species have been shown to have bioactive metabolites. The WHO reported that around 21,000 plant species contained medicinal properties (*Borah & Banik, 2020*).

Dill (*Anethum graveolens* L.) belongs to the family Apiaceae and is one of the most popular dietary herbs in the world. This aromatic plant originated in the Mediterranean and Southwest Asia (*Bailer et al., 2001*; *Singh et al., 2005*), and its biological and pharmacological properties have been studied. Some researchers have reported that dill exhibited various phytochemicals in leaves, stems, and roots that are rich in tannins, terpenoids, cardiac glycosides, and flavonoids (*Jana & Shekhawat, 2010*). Previous studies have mainly focused on dill seeds. Oil derived from dill seeds had many medicinal properties, such as antioxidant, antimicrobial (*Hadi et al., 2024*), antiproliferative, and anticancer properties (*Al-Sheddi et al., 2019*). Ethanol-extracted dill seed exhibited notable anticancer and intracellular antioxidant properties (*Al-Oqail & Farshori, 2021*). Additionally, the ethyl acetate extract revealed antiproliferative effects (*El Mansouri et al., 2014*). In addition, dried dill leaf powder had antidiabetic properties (*Haidari et al., 2020*). The antioxidant activities of dill leaves and flowers were evidenced in extracts from water, acetone, ethanol, ethyl acetate, and hexane (*Isbilir & Sagiroglu, 2011*; *Shyu et al., 2009*). Interestingly, the study of the delayed process of skin aging showed that dill may improve the elasticity of dermis equivalents (DEs) *in vitro* as well as skin biomechanical properties and appearance when dill extract is used as one of the formulations *in vivo* (*Sohm et al., 2011*). Dill can suppress reactive oxygen species (ROS), including hydroxyl radicals ($\cdot$OH), superoxide anion ($O_2^{\cdot-}$), hydrogen peroxide ($H_2O_2$), and singlet oxygen ($1O_2$) (*Hajimohammadi & Verjani, 2019*).

A metabolomic approach has been employed to investigate bioactive metabolite components in plants, owing to its effectiveness in analytical profiling platforms (*Salem et al., 2020*). Nevertheless, there are currently no reports on the impact of extraction temperature and duration on the metabolite composition of aqueous-extracted dill leaves

at 27 °C compared to 90 °C as a hot water model, aiming to replicate the tea brew temperature. The study aimed to investigate the metabolic profiles of aqueous-extracted dill leaves under various conditions using untargeted $^1$H NMR analysis. Antioxidant properties of dill leaf extracts were assessed, and their longevity effects were investigated using primary human dermal fibroblasts (HDFs) as a model.

## MATERIALS & METHODS

### Plant collection
Dill leaves were acquired from a local market in Lom Sak District of Phetchabun Province, Thailand, where they were cultivated in March 2023. The dried dill leaves were then ground into a fine powder using an electrical blender (SHARP blender EMC-15, Japan).

### Chemicals and reagents
Folin-Ciocalteau, 2,4,6-Tripyridyl-s-triazine (TPTZ), (±)-6-Hydroxy-2,5,7,8-tetramethylchromane-2-carboxylic acid (Trolox), gallic acid, ABTS (2,2′-azino-bis-(3-ethylbenzothiazoline-6-sulfonic acid)), and Sulforhodamine B were purchased from Sigma-Aldrich (St. Louis, MO, USA). 2,2′-Azobis(2-amidinopropane) dihydrochloride (AAPH) was purchased from Acros Organics (Geel, Belgium). Enhanced chemiluminescence plus solution (ECL) was provided by GE Healthcare (Chicago, IL, USA). Pierce bicinchoninic acid (BCA) protein assay kit and bovine serum albumin (BSA) were ordered from Thermo Scientific (Waltham, MA, USA). Potassium dihydrogen phosphate ($KH_2PO_4$) and deuterium oxide ($D_2O$) were purchased from Merck (Darmstadt, Germany). Penicillin-streptomycin and trypsin-EDTA were obtained from Life Technologies (Grand Island, NY, USA). The internal standard reference for NMR analysis, 3-(Trimethyl-silyl) propionic acid-d4 sodium salt (TSP), was obtained from Cambridge Isotope Laboratories (Andover, MA, USA).

The primary antibodies (Ab) including mouse anti-mouse β-actin (A5441), anti-rabbit SIRT6 (AB191385), anti-rabbit FOXO3 (AB109629), anti-AMPK (5831S), anti-Akt (4685S), and anti-mTOR (AB32028), were purchased from Abcam (Cambridge, UK). The secondary antibodies (anti-mouse antibody and anti-rabbit antibody) were obtained from Thermo Fisher Scientific (Waltham, MA, USA).

### Cell culture
Primary human dermal fibroblasts (HDFs) (Catalogue number ATCC® PCS-201-012™, lot number 70015617, 39-year-old African-American male) were an adult skin cell line purchased from American Type Culture Collection (ATCC) (Manassas, VA, USA). Cells were cultured in a DMEM nutrient mixture supplemented with 10% heat-inactivated fetal bovine serum (Thermo Fisher Scientific, Waltham, MA, USA), 100 U/mL penicillin, and 100 μg/mL streptomycin at 37 °C in a humidified incubator containing a 5% $CO_2$ as previously described (*Nakorn et al., 2024*).

### Aqueous extract preparation
Fifty grams of dill leaf powder was extracted by macerating in 200 mL of distilled water under two different conditions, 27 °C and 90 °C, with extraction times of 2 min, 1 h, and

2 h. Afterward, all samples were filtered through Whatman® No. 1 filter paper, and the extracts were collected upon removing water using a lyophilizer (Labconco, Kansas City, MO, USA). All conditions were prepared in five replicates. Extracts were stored at 4 °C with light protection for further analysis (*Cheng, Xue & Yang, 2023*).

## Quantification of antioxidant activities

In accordance with the mechanisms of antioxidant substances, *in vitro* antioxidant approaches may be categorized into two primary groups: (1) single electron transfer (SET) and (2) hydrogen atom transfer (HAT) reactions (*Santos-Sánchez et al., 2019*). The study included four distinct antioxidant tests to thoroughly evaluate the antioxidant properties of aqueous dill leaf extracts. This study included (1) total phenolic content (TPC) measured by the Folin-Ciocalteu method and the FRAP test, which demonstrates a single electron transfer (SET) reaction. (2) The hydrogen atom transfer (HAT) reaction was shown in the ORAC test. (3) The ABTS assay identified both SET and HAT reactions.

## Determination of total phenolic content

The total phenolic content (TPC) was determined using the Folin-Ciocalteu method (*Singleton, Orthofer & Lamuela-Raventós, 1999*) with a few modifications. One hundred microliters of 0.2 N Folin-Ciocalteu reagent was mixed with 1,000 $\mu$g/mL of dill extracts. One hundred microliters of 7% $Na_2CO_3$ was added after 30 min of incubation with light protection. Absorbance was then measured at 765 nm using a microplate reader (Berthold Technologies, Oak Ridge, TN, USA). Experiments were performed in triplicate. TPC was measured in micrograms of gallic acid equivalents per milligram of dry weight ($\mu$g GAE/mg DW), compared to the reference compound.

## Antioxidant assessments by a ferric-reducing antioxidant power assay

The ferric-reducing antioxidant power (FRAP) reagent was freshly prepared by mixing 300 mM sodium acetate buffer (pH 3.6), 20 mM $FeCl_3 \cdot 6H_2O$ in 40 mM HCl, and 10 mM TPTZ in a ratio of 10:1:1. Next, 100 $\mu$L of the FRAP reagent was mixed with 100 $\mu$L of dill leaf extract (1,000 $\mu$g/mL) and allowed to stand at room temperature for 5 min. A microplate reader (Berthold Technologies, Oak Ridge, TN, USA) was then used to measure the absorbance at 593 nm. The antioxidant capacity was given as micrograms of gallic acid equivalents (GAE) per milligram of dry weight ($\mu$g GAE/mg DW) (*Benzie & Strain, 1996*).

## ABTS (2,2′-azino-bis-(3-ethylbenzothiazoline-6-sulfonic acid)) assay

ABTS or Trolox equivalent antioxidant capacity (TEAC) was determined as follows: ABTS·$^+$ was generated by reacting 7 mM aqueous ABTS with 2.45 mM potassium persulfate ($K_2S_2O_8$) in the dark at room temperature for 12 to 16 h until a dark blue color developed. Thereafter, the absorbance of the ABTS solution was adjusted to 0.70 ± 0.02 at 734 nm. Consequently, 10 $\mu$L of dill leaf extract (1,000 $\mu$g/mL) was mixed with 195 $\mu$L of the ABTS radical solution. The reaction mixture was incubated for 30 min, and the absorbance was measured at 734 nm in triplicate using a microplate reader (Berthold Technologies,

Oak Ridge, TN, USA). Finally, the antioxidant capacity was expressed as micrograms of Trolox® equivalents (TE) per milligram of dry weight (µg TE/mg DW) (*Re et al., 1999*).

## Determination of oxygen radical absorbance capacity assay

Twenty-five µL of dill leaf extract (1,000 µg/mL) was prepared in 10 mM phosphate buffer (pH 7.4) in a microplate. Then, 150 µL of 10 nM fluorescein was added. The microplate was sealed and incubated for 30 min at 37 °C. Next, 25 µL of 240 mM AAPH reagent was added, and the measurement was taken using a fluorescent microplate reader (Berthold Technologies, Oak Ridge, TN, USA) at an excitation wavelength of 485 nm and an emission wavelength of 520 nm. This assay used Trolox as a standard and reported result in micrograms of Trolox® equivalents (TE) per milligram of dry weight (µg TE/mg DW) (*Gillespie, Chae & Ainsworth, 2007*).

## $^1$H NMR-based metabolomics analysis of medicinal plants

The chemical constituents of the dill water extracts were determined using proton nuclear magnetic resonance spectroscopy ($^1$H NMR). A total of 100 mg of plant extract was added to a 1.5 mL microcentrifuge tube. Then, one mL of $D_2O$ (pH 7.4) containing 1.5 M $KH_2PO_4$, 2 mM $NaN_3$, and 0.1% (w/v) TSP was added and vortexed for 1 min at room temperature. The mixture was subjected to ultrasonication for 30 min twice at room temperature, after which it was centrifuged at 15,000 rpm at 4 °C for 15 min (*Promraksa et al., 2019*). A total of 600 µL of the supernatant was transferred to an NMR tube, and the $^1$H NMR spectra were obtained using a 400 MHz $^1$H NMR spectrometer (Bruker, Billerica, MA, USA). The Carr-Purcell-Meiboom-Gill (CPMG) sequence was utilized as a presaturation pulse sequence for water suppression, which was derived from the standard pulse sequence in the spectrometer library. A total of 64 scans were collected into 32 K data points with a relaxation delay of 4 s. Subsequently, chemical shift referencing, phasing, and baseline correction were performed. We then used MATLAB for spectral alignment, binning (or bucketing), normalization, and scaling. Multivariate statistical analysis was used for identifying all different samples. PCA and OPLS-DA analyses were performed using MetaboAnalyst 6.0 (University of Alberta, Canada). The data was Pareto (Par) scaled. Next, we used the Statistical Total Correlation Spectroscopy (STOCSY) and Subset Optimization by Reference Matching (STORM) techniques. Furthermore, the resonances of interest were searched against online metabolite databases, such as the Human Metabolome Database (HMDB). To further support the identification of metabolites, we carried out a 2D $^1$H NMR investigation, specifically *J*-resolved (JRES) spectroscopy, with Topspin 4.3.0 software. Additionally, NMR can provide quantification of absolute concentrations. The maximum intensity of the metabolites of interest was quantified in relation to the known standard reference concentration (TSP) to quantify the absolute concentration using the formula below (*Phetcharaburanin et al., 2020*).

$$\text{Absolute concentration of } X = \frac{N \text{ of protons of TSP}}{N \text{ of protons of } X} \times \frac{\text{Maximum intensity of peak } X}{\text{Maximum intensity of peak TSP}}$$
$$\times \text{Concentration of TSP}$$

## Detection of intracellular reactive oxygen species

Briefly, human dermal fibroblast cells (2,500 cells/well) were plated in black wall 96-well plates in DMEM media for 18 h. The cells were then pretreated with 1,000 µg/mL of dill leaf extracts, which had been extracted at 27 °C and 90 °C for 2 min, in DMEM media for 48 h. Ten µg/mL of gallic acid was used as a positive control. Afterward, the cells were rinsed twice with PBS and incubated with 10 mM of $H_2O_2$, with or without the dill leaf extracts and gallic acid, in DMEM media. Following this, the cells were washed with PBS. To determine intracellular ROS production, the cells were incubated with 5 µM of CM-$H_2$DCFDA prepared in culture medium for 45 min. The CM-$H_2$DCFDA-containing media was then removed and replaced with 100 µL of PBS. Finally, the fluorescence intensity was measured using a microplate reader (Berthold Technologies, Oak Ridge, TN, USA). The experiments were performed in triplicate (*Nakorn et al., 2024*).

## Western blot analysis

Human dermal fibroblasts (HDFs) ($2 \times 10^5$ cells/mL) were subcultured until they reached 70% confluence and treated with 1,000 µg/mL of dill leaf extracts for 96 h. Following incubation, cell pellets were collected and lysed using NP40 lysis buffer containing 0.1% sodium dodecyl sulfate (SDS), 0.5% sodium deoxycholate, 50 mM Tris, 1% Tween 20, and protease cocktail inhibitor (Roche, Basel, Switzerland). The amount of total protein was measured using a BCA protein assay kit (Thermo Fisher Scientific, Waltham, MA, USA). Thirty µg of protein lysates were resolved in 4X SDS buffer and boiled at 95 °C for 5 min. Next, the protein mixture was loaded, separated under 10% w/v SDS-polyacrylamide gel electrophoresis, and transferred to a polyvinylidene fluoride (PVDF) membrane (Merck, Rahway, NJ, USA). The membrane was treated with 5% w/v skim milk in Tris-buffered saline (TBS) at room temperature for 1 h to block non-specific protein. The membrane was subsequently incubated with primary antibodies, including mTOR, Akt, AMPK, FOXO3, or SIRT6, at 4 °C overnight. The membrane was then washed with TBS containing 0.1% Tween 20 and probed with secondary antibodies conjugated to horseradish peroxidase at room temperature for 1 h. After rinsing with TBS-T, the membranes were exposed to the ECL Prime Western Blotting Detection Reagent (Amersham, Cytiva Life Sciences, UK). The immunoblot was analyzed using ImageJ (National Institutes of Health, Bethesda, MD, USA) (*Nakorn et al., 2024*).

## Statistical analysis

All spectra were manually chemical shift referenced, phased, and baseline corrected in TopSpin 4.3.0 (Bruker BioSpin, Rheinstetten, Germany). Data processing included spectral alignment, binning or bucketing, normalization, and Pareto scaling. The Statistical Total Correlation Spectroscopy (STOCSY) and Subset Optimization by Reference Matching
(STORM) tools performed in MATLAB (R2022a) (MathWorks, Natick, MA, USA) were applied to confirm the assignment of correlated resonances. 2D $^1$H NMR investigation, *J*-resolved (JRES), was carried out to further support the metabolites of identification in Topspin 4.3.0 software (Bruker BioSpin, Rheinstetten, Germany). Data modeling and statistical analysis were executed utilizing a principal component analysis (PCA) and orthogonal partial least squares for discriminant analysis (O-PLS-DA); box plots and heatmaps were generated using MetaboAnalyst 6.0 (University of Alberta, Alberta, Canada). Antioxidant screening results are presented as mean $\pm$ standard deviation (SD) from three independent experiments. One-way ANOVA was conducted using GraphPad Prism software version 8.0.1 (GraphPad Software, San Diego, CA, USA). A *p*-value $<0.05$ was considered statistically significant.

# RESULTS

## Percentage yield of dill leaf extracts

Dill leaf powder was extracted with water, which was categorized into two main groups: 27 °C and 90 °C water extractions. For 27 °C, the water extractions at 2 min, 1 h, and 2 h exhibited average yields of 13.52%, 14.04%, and 14.68%, respectively. The 90 °C water extractions at 2 min, 1 h, and 2 h showed average yields of 14.42%, 14.11%, and 13.90%, respectively. The results indicated that the average yields of dill leaf extracts obtained from 27 °C and 90 °C water extractions were not different (Table S1).

## Metabolic profiles of dill leaf extracts

The metabolic profiles of dill leaf extracts were analyzed using $^1$H NMR spectroscopy at 400 MHz in $D_2O$. The spectral data were subsequently acquired and preprocessed. To investigate patterns, correlations, clusters, and outliers in the dill samples, principal component analysis (PCA) plots were generated. The PCA score plot indicated a tight clustering of quality control (QC) samples, reflecting minimal analytical variation and high precision (Fig. 1A). The results of the PCA score plot demonstrated that all classes, including six conditions in the 27 °C and 90 °C water extraction groups of dill leaf extracts, could be distinguished along the first principal component (PC1 = 81.1% for all conditions, PC1 = 84.3% for 27 °C, and PC1 = 47.3% for 90 °C) (Figs. 1B–1D). Notably, 27 °C and 90 °C conditions were clearly separated (Fig. 1B). In the 27 °C water conditions, the model at room temperature (27 °C) showed that dill leaf extracts for 2 min and 1 h were clearly different from each other according to PC1 (Fig. 1C). In contrast, the hot water (90 °C) model showed less clear separation among the groups (Fig. 1D).

Nine pairwise OPLS-DA models were constructed to investigate the effects of temperature and duration on the chemical compositions of dill leaf extracts, distinguishing metabolite profiles among aqueous-extracted dill leaves from different conditions (*Wang et al., 2022*). At different extraction temperatures but at the same time points (Figs. 2A–2C), OPLS-DA exhibited that temperature affects the metabolite profiles of dill. Furthermore, Figs. 2D–2F showed the effects of extraction duration on dill metabolite profiles under 27 °C water conditions. Additionally, three pairwise OPLS-DA plots (Figs. 2G–2I) visualized the effects of extraction duration on the metabolite profiles of dill leaf extracts under 90 °C

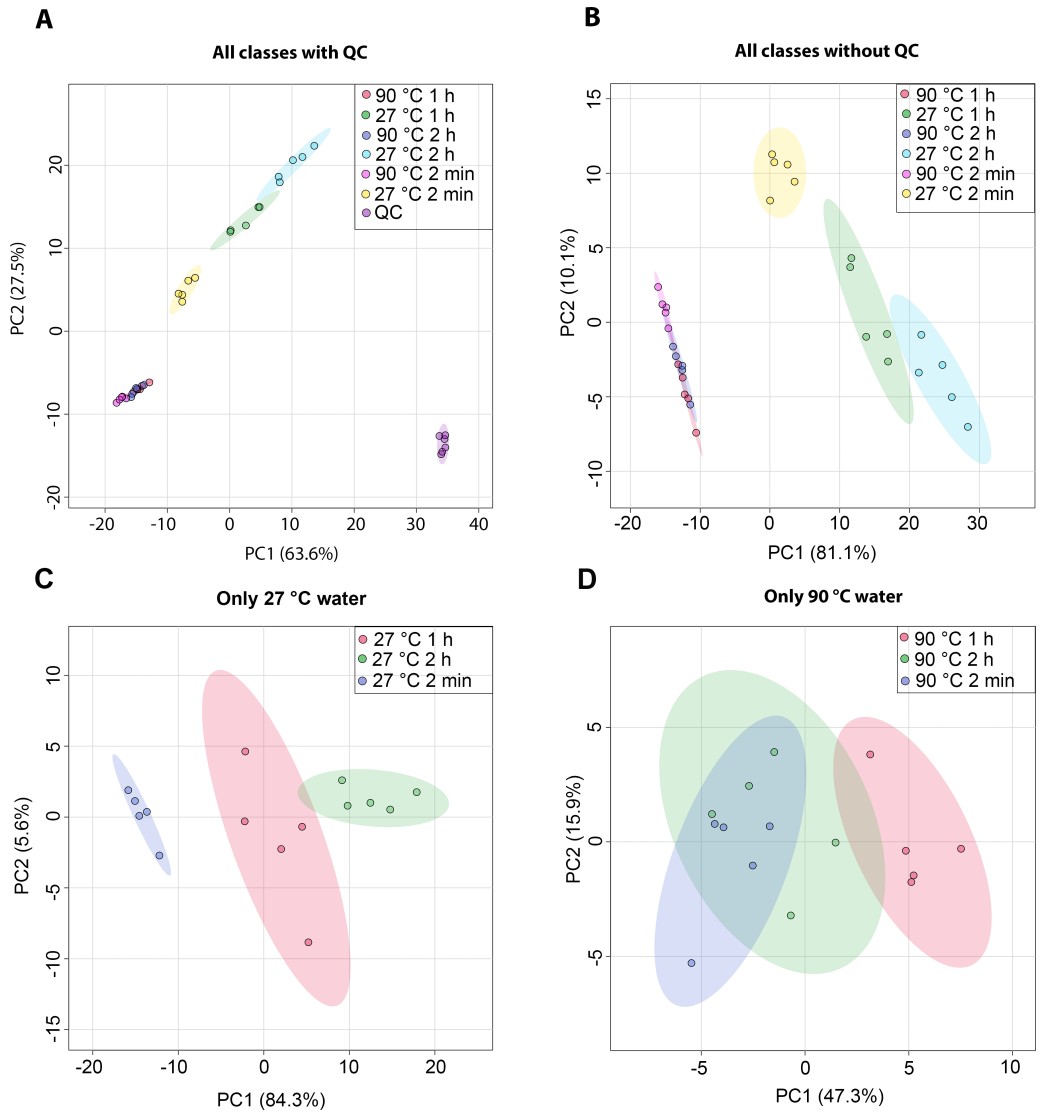

**Figure 1** **Overview of the metabolic similarities and differences observed in various conditions of dill leaf extracts.** PCA score plots derived from $^1$H NMR data show: (A) all classes including quality control (QC) samples, (B) all classes excluding QC samples, (C) 27 °C water conditions, and (D) 90 °C water conditions.

water conditions (Figs. 2G–2I). These findings suggest that both temperature and extraction duration significantly influenced the changes in the metabolic profiles of dill leaf extracts according to their concentrations.

Before data analysis, we managed the preprocessing of NMR data, which included chemical shift referencing, baseline correction, phasing, peak alignment, normalization, and scaling. In the $^1$H NMR, metabolites were identified through assignments of correlated resonances using Statistical Total Correlation Spectroscopy (STOCSY) and Subset Optimization by Reference Matching (STORM). These metabolites from dill extracts were then compared with online databases, such as Chenomx software, the Biological Magnetic

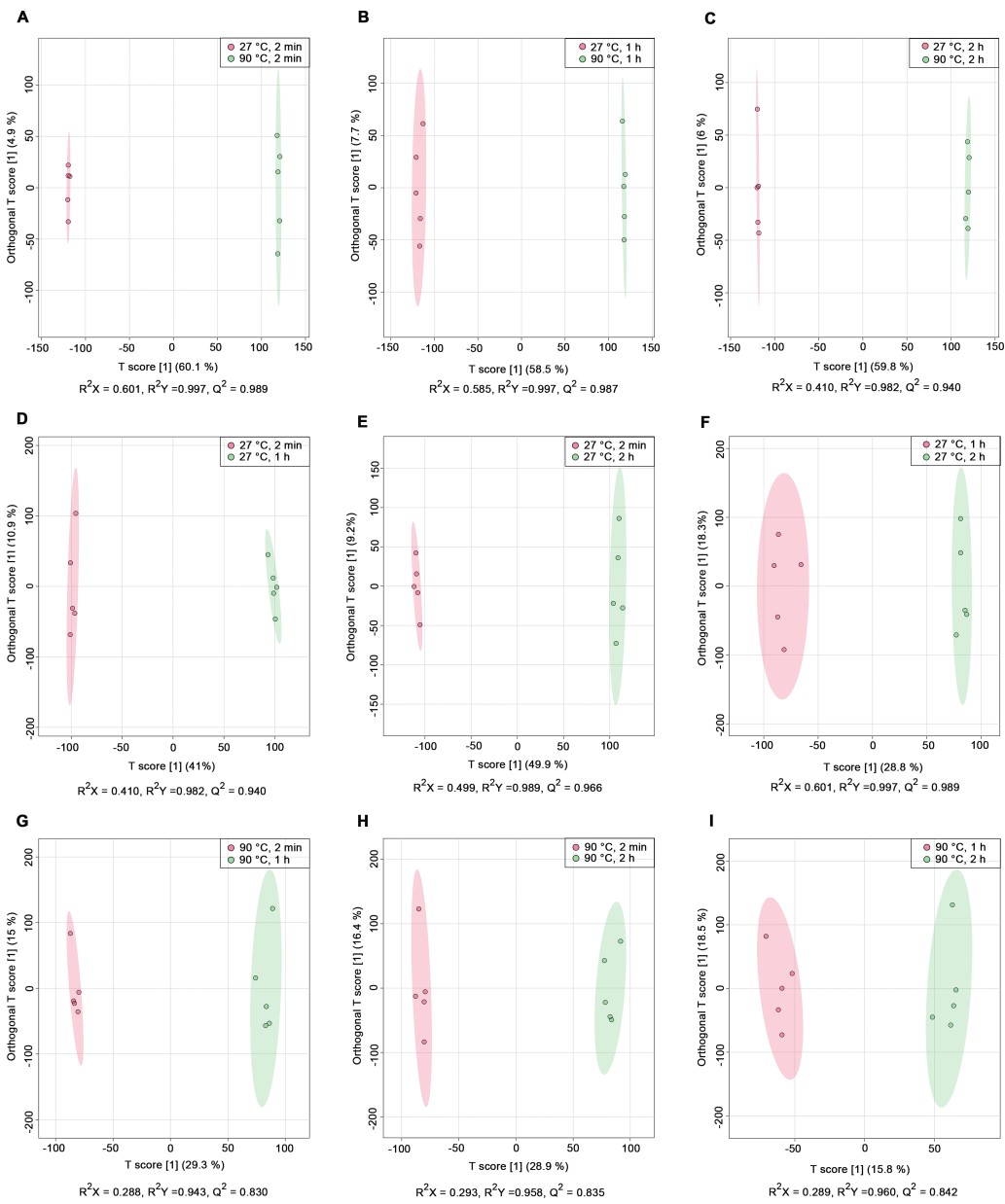

**Figure 2** **O-PLS-DA plots illustrating pairwise comparisons between dill leaf extracts under different conditions to assess the effects of temperature and extraction duration on their chemical compositions.** (A) 27 °C, 2 min *vs.* 90 °C, 2 min; (B) 27 °C, 1 h *vs.* 90 °C, 1 h; (C) 27 °C, 2 h *vs.* 90 °C, 2 h; (D) 27 °C, 2 min *vs.* 27 °C, 1 h; (E) 27 °C, 2 min *vs.* 27 °C, 2 h; (F) 27 °C, 1 h *vs.* 27 °C, 2 h; (G) 90 °C, 2 min *vs.* 90 °C, 1 h; (H) 90 °C, 2 min *vs.* 90 °C, 2 h; and (I) 90 °C, 1 h *vs.* 90 °C, 2 h.

Resonance Data Bank (BMRB), and the Human Metabolome Database (HMDB). Out of the 69 signals found in the spectra, we identified 45 possible metabolites using 1D $^1$H NMR analysis. Due to the complexity of the spectra and overlapping peaks, we employed 2D *J*-resolved $^1$H NMR to analyze the various types of metabolite peaks. Conclusively, 41 metabolites were confirmed, as shown in Table 1, while 28 metabolites remain unidentified,
as indicated in Table S2. Additionally, representative median $^1$H NMR spectra of dill leaf extracts under six conditions are shown in Fig. S1.

Absolute concentrations were quantified using the maximum intensity of all identified metabolites, with TSP as the internal standard at a known concentration (Table S3). A one-way ANOVA test ($p$-value < 0.05) was done to obtain significant metabolites based on their concentrations. Among the 41 metabolites identified in dill extracts, five metabolites, including folate, pyridoxal, inosine, adenine, and indole-3-lactate, showed significantly elevated concentration levels under 90 °C conditions (Fig. 3A). In contrast, cellobiose, α-glucose, and β-glucose exhibited significantly higher concentration levels under 27 °C water conditions (Fig. 3B).

### Antioxidant activities of aqueous-extracted dill leaves under various conditions

We determined the antioxidant capacities of dill leaf extracts (1,000 mg/mL) using four methods, namely, TPC, FRAP, ABTS, and ORAC assays. In dill leaf extracts at 27 °C, the total phenolic content was around 24–27 µg gallic acid equivalents (GAE) per mg dry weight (DW), with a tendency to diminish over time (Fig. 4A). In contrast, the extracts at 90 °C had a total phenolic content of approximately 26–27 µg GAE/mg DW, and there was no significant change over time (Fig. 4B). The FRAP assay indicated that water extracts at 27 °C had an antioxidant capacity of around 11–12 µg GAE/mg DW, which also diminished over time, comparable to the TPC assay (Fig. 4D). The antioxidant capacity in water extracts at 90 °C was approximately 14–15 µg GAE/mg DW, with no significant variations noted across different time points (Fig. 4E). The ABTS experiment indicated that water extracts at 27 °C exhibited an antioxidant capacity ranging from 12 to 26 µg Trolox equivalents (TE)/mg DW, with minor variations across the groups (Fig. 4G). On the contrary, 90 °C water conditions displayed antioxidant capacity levels at around 19–38 µg TE/mg DW (Fig. 4H). The ORAC test indicated that the antioxidant levels were similar for the 27 °C and 90 °C water extracts, measuring about 21–22 and 24–25 µg TE/mg DW, respectively (Figs. 4J–4K). In conclusion, across all four antioxidant assays, a comparison of antioxidant capacity between 27 °C and 90 °C water conditions at equivalent time intervals of 2 min, 1 h, and 2 h revealed that the extracts from 90 °C water exhibited significantly greater antioxidant capacity than those from 27 °C water (Figs. 4C, 4F, 4I, and 4L).

### Correlation analysis of metabolites and antioxidant activities

A Pearson correlation analysis was performed to investigate the relationship between significant metabolites and antioxidant capacity across different conditions, specifically at 27 °C (2 min, 1 h, and 2 h) and 90 °C (2 min, 1 h, and 2 h), as depicted in the heatmaps (Fig. 5). The study revealed distinct connection patterns, highlighting significant associations between certain metabolites and antioxidant tests (FRAP, ABTS, and ORAC). Pyridoxal, indole-3-lactate, adenine, inosine, and folate exhibited substantial positive correlations with all three antioxidant experiments, suggesting their possible role in enhancing antioxidant capacity. In contrast, β-hydroxybutyrate, cellobiose, and α-glucose exhibited quite moderate negative associations with antioxidant assays.

Thanuma et al. (2025), PeerJ, DOI 10.7717/peerj.19567

**Table 1   List of all metabolites found in NMR spectra of dill leaf extracts.**

| No. | Metabolite | Chemical class | Chemical subclass | Chemical shift value | HMDB ID |
|-----|-----------|----------------|-------------------|----------------------|---------|
| 1 | Pantothenate | Organooxygen compounds | Alcohols and polyols | 0.93(s), 0.95(s), 2.42(t), 3.43(q), 3.58(d), 3.99(s), 4.14(s), 8.01(s) | HMDB0000210 |
| 2 | Leucine | Carboxylic acids and derivatives | Amino acids, peptides, and analogues | 0.96(m), 1.72(m), 3.73(m) | HMDB0000687 |
| 3 | Isoleucine | Carboxylic acids and derivatives | Amino acids, peptides, and analogues | 1.00(d), 1.26(m), 1.48(m), 1.98(m), 3.68(d) | HMDB0000172 |
| 4 | Valine | Carboxylic acids and derivatives | Amino acids, peptides, and analogues | 0.99(d), 2.28(m), 3.60(d) | HMDB0000883 |
| 5 | α-ketoisovalerate | Keto acids and derivatives | Short-chain keto acids and derivatives | 1.14(d), 3.02(m) | HMDB0000019 |
| 6 | β-hydroxybutyrate | Hydroxy acids and derivatives | Beta hydroxy acids and derivatives | 1.18(d), 2.32(dd), 2.41(dd), 4.18(m) | HMDB0000011 |
| 7 | (S)-3-hydroxybutyrate | Hydroxy acids and derivatives | Beta hydroxy acids and derivatives | 1.23(d), 2.44(d), 4.19(m) | HMDB0000442 |
| 8 | Threonine | Carboxylic acids and derivatives | Amino acids, peptides, and analogues | 1.34(d), 3.58(d), 4.30(m) | HMDB0000167 |
| 9 | Lactate | Hydroxy acids and derivatives | Alpha hydroxy acids and derivatives | 1.34(d), 4.12(q) | HMDB0000190 |
| 10 | Alanine | Carboxylic acids and derivatives | Amino acids, peptides, and analogues | 1.50(d), 3.79(q) | HMDB0001310 |
| 11 | Acetate | Carboxylic acids and derivatives | Carboxylic acids | 1.95(s) | HMDB0000042 |
| 12 | α-hydroxyisovalerate | Fatty Acyls | Fatty acids and conjugates | 0.84(d), 0.97(d), 2.00(m), 3.86(d) | HMDB0000407 |
| 13 | Isovalerate | Fatty Acyls | Fatty acids and conjugates | 0.89(d), 1.94(m), 2.05(d) | HMDB0000718 |
| 14 | α-ketoisocaproate | Keto acids and derivatives | Short-chain keto acids and derivatives | 0.90(d), 2.08(m), 2.60(d) | HMDB0000695 |
| 15 | Homocysteine | Carboxylic acids and derivatives | Amino acids, peptides, and analogues | 2.15(m), 2.28(m), 2.67(m), 2.84(t), 3.87(t) | HMDB0000742 |
| 16 | Methionine | Carboxylic acids and derivatives | Amino acids, peptides, and analogues | 2.15(s), 2.16(m), 2.65(t), 3.85(t) | HMDB0000696 |
| 17 | γ-aminobutyrate | Carboxylic acids and derivatives | Amino acids, peptides, and analogues | 1.92(q), 2.30(t), 3.02(t) | HMDB0000112 |
| 18 | Succinate | Carboxylic acids and derivatives | Dicarboxylic acids and derivatives | 2.42(s) | HMDB0000254 |

Thanuma et al. (2025), *PeerJ*, DOI 10.7717/peerj.19567

**Table 1** (*continued*)

| No. | Metabolite | Chemical class | Chemical subclass | Chemical shift value | HMDB ID |
|---|---|---|---|---|---|
| 19 | Pyridoxamine | Pyridines and derivatives | Pyridoxamines | 2.44(s), 4.32(s), 7.70(s) | HMDB0001431 |
| 20 | Citrate | Carboxylic acids and derivatives | Tricarboxylic acids and derivatives | 2.52(d), 2.69(d) | HMDB0000094 |
| 21 | Aspartate | Carboxylic acids and derivatives | Amino acids, peptides, and analogues | 2.67(dd), 2.77(dd), 3.87(dd) | HMDB0000191 |
| 22 | Sarcosine | Carboxylic acids and derivatives | Amino acids, peptides, and analogues | 2.74(s), 3.60(s) | HMDB0000271 |
| 23 | Acetylcholine | Organonitrogen compounds | Quaternary ammonium salts | 2.15(s), 3.22(s), 3.73(m) | HMDB0000895 |
| 24 | 3,7-dimethylurate | Imidazopyrimidines | Purines and purine derivatives | 3.34(s), 3.36(s) | HMDB0001982 |
| 25 | Proline | Carboxylic acids and derivatives | Amino acids, peptides, and analogues | 1.92(m), 2.29(m), 3.36(m), 4.12(dd) | HMDB0251528 |
| 26 | Sucrose | Organooxygen compounds | Carbohydrates and carbohydrate conjugates | 3.53(q), 4.24(d), 5.43(d) | HMDB0000258 |
| 27 | Malate | Hydroxy acids and derivatives | Beta hydroxy acids and derivatives | 2.36(dd), 2.67(dd), 4.30(dd) | HMDB0000156 |
| 28 | $\beta$-glucose | Organooxygen compounds | Carbohydrates and carbohydrate conjugates | 3.30(dd), 3.44(m), 3.73(m), 4.63(d), 5.27(d) | HMDB0000122 |
| 29 | Cellobiose | Organooxygen compounds | Carbohydrates and carbohydrate conjugates | 3.26(m), 3.40(m), 3.89(m), 4.68(d), 5.53(d) | HMDB0000055 |
| 30 | $\alpha$-glucose | Organooxygen compounds | Carbohydrates and carbohydrate conjugates | 3.28(dd), 3.38(m), 3.71(m), 3.81(m), 3.87(dd), 4.61(d), 5.28(d) | HMDB0003345 |
| 31 | Uracil | Diazines | Pyrimidines and pyrimidine derivatives | 5.81(d), 7.56(d) | HMDB0000300 |
| 32 | Fumarate | Carboxylic acids and derivatives | Dicarboxylic acids and derivatives | 6.54(s) | HMDB0000134 |
| 33 | Tyrosine | Carboxylic acids and derivatives | Amino acids, peptides, and analogues | 3.06(dd), 3.30(dd), 3.94(dd), 6.92(d), 7.19(d) | HMDB0000158 |
**Table 1** (*continued*)

| No. | Metabolite | Chemical class | Chemical subclass | Chemical shift value | HMDB ID |
|---|---|---|---|---|---|
| 34 | Phenylalanine | Carboxylic acids and derivatives | Amino acids, peptides, and analogues | 3.14(dd), 3.30(dd), 4.00(dd), 7.32(m) | HMDB0000159 |
| 35 | Tryptophan | Indoles and derivatives | Indolyl carboxylic acids and derivatives | 3.30(dd), 3.49(dd), 4.05(dd), 7.33(s), 7.59(d), 7.74(d) | HMDB0000929 |
| 36 | Pyridoxal | Pyridines and derivatives | Pyridine carboxaldehydes | 2.49(s), 5.16(dd), 6.54(s), 7.56(s) | HMDB0001545 |
| 37 | Indole-3-lactate | Indoles and derivatives | Indolyl carboxylic acids and derivatives | 3.07(dd), 3.28(d), 3.67(s), 4.36(m), 7.19(t), 7.25(t), 7.27(t), 7.56(d), 7.74(d) | HMDB0000671 |
| 38 | Adenine | Imidazopyrimidines | Purines and purine derivatives | 8.18(s) | HMDB0000034 |
| 39 | Inosine | Purine nucleosides | Not available | 3.82(dd), 3.92(dd), 4.33(q), 4.37(t), 6.05(d), 8.25(s), 8.34(s) | HMDB0000195 |
| 40 | Formate | Carboxylic acids and derivatives | Carboxylic acids | 8.49(s) | HMDB0000142 |
| 41 | Folate | Phenols | Methoxyphenols | 2.00(m), 2.12(m), 2.32(m), 4.30(m), 4.54(s), 6.72(d), 7.70(d), 8.05(d), 8.60(s) | HMDB0000121 |
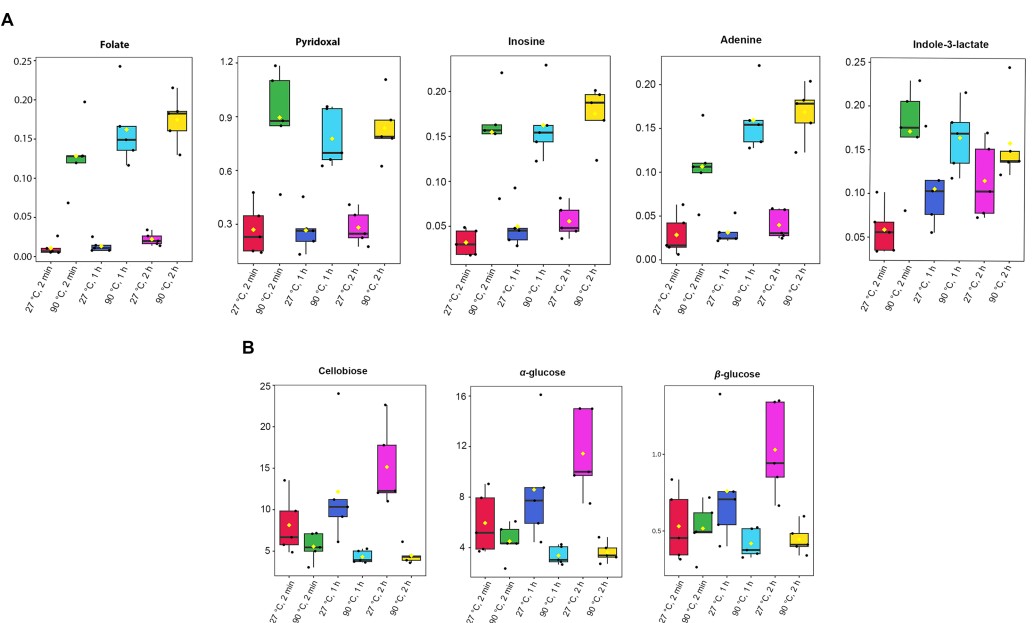

**Figure 3  Box plots of concentrations of eight significant metabolites in dill leaf extracts under different conditions.** (A) five metabolites, including folate, pyridoxal, inosine, adenine, and indole-3-lactate, showed significantly elevated concentration levels under 90 °C conditions (B) three metabolites, including cellobiose, α-glucose, and β-glucose exhibited significantly higher concentration levels under 27 °C water conditions, respectively. Data are presented as mean ± SD ($n = 5$). The significant difference was determined using one-way ANOVA test ($p < 0.05$).

## Intracellular reactive oxygen species in HDFs following treatment with dill leaf extracts

The proportion of cytosolic oxidation in HDFs was quantified using the CM-H$_2$DCFDA method following treatment with dill leaf extracts, which identifies intracellular reactive oxygen species (ROS). The experiments were conducted three times independently. The results showed that dill leaf extracts significantly decreased ($p < 0.05$) the cytosolic oxidation caused by H$_2$O$_2$ in HDFs. Furthermore, no significant differences in antioxidant activity were observed between 27 °C and 90 °C water conditions, as shown in Fig. 6.

## Altered levels of longevity-associated proteins in dill leaf extract-treated HDFs

The HDFs were treated with dill leaf extracts to investigate the effect of the extracts on the expression of longevity-related proteins, including AMPK, Akt, mTOR, SIRT6, and FOXO3 (Fig. 7). The results showed the expression of mTOR and Akt significantly decreased in longevity pathways involving autophagy, while AMPK and FOXO3 proteins increased in response to treatment with dill leaf extract. Moreover, HDFs treated with dill leaf extracts exhibited elevated expression of SIRT6 protein, which is associated with longevity *via* the DNA repair pathway.

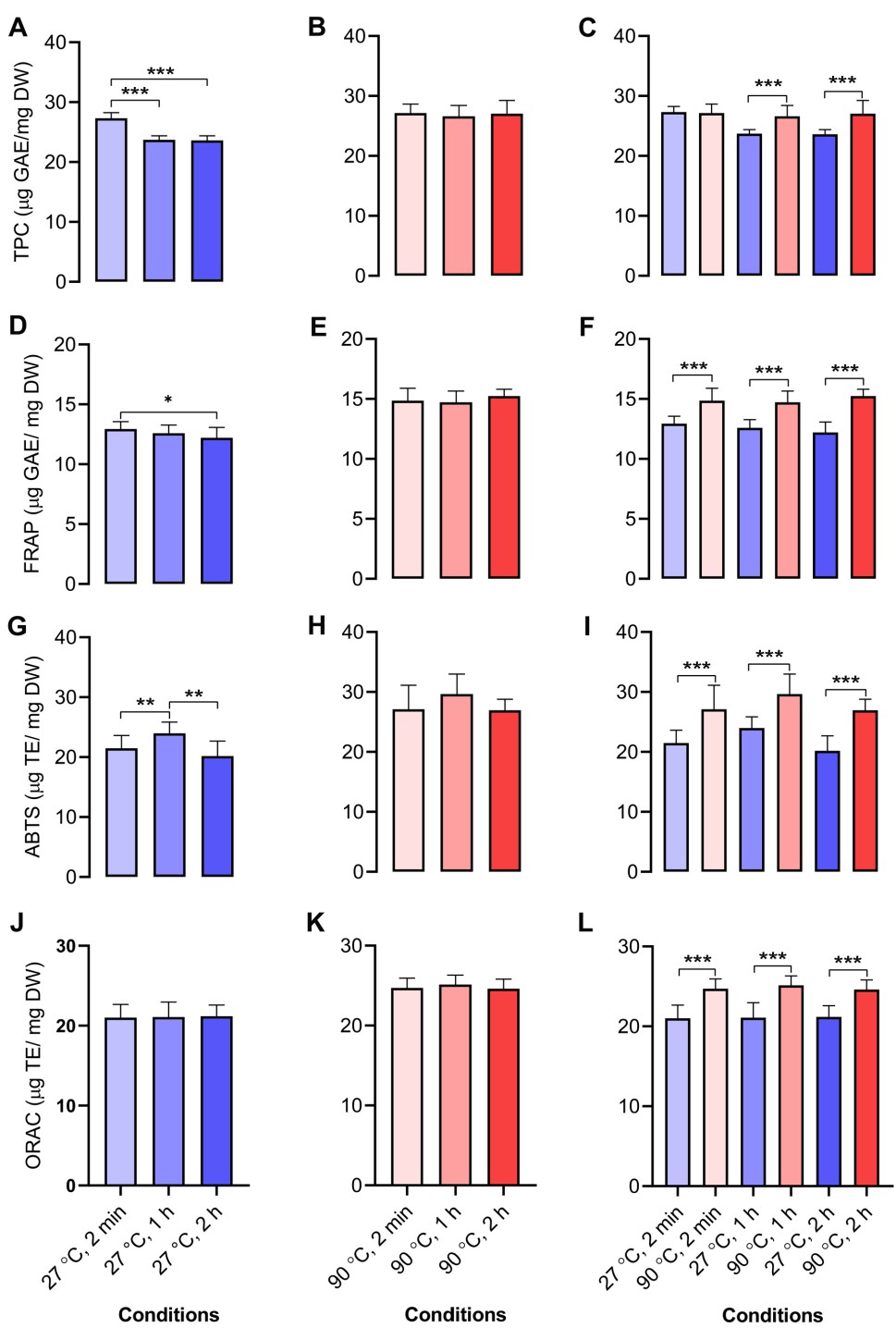

**Figure 4  Antioxidant activities of aqueous-extracted dill leaves under various conditions.** Antioxidant capacities of dill leaf extracts (1,000 m g/mL) were determined using four methods, including TPC, FRAP, ABTS, and ORAC assays. TPC assay: (A) 27 °C water; (continued on next page...)

**Figure 4 (…continued)**
(B) 90 °C water; (C) 27 °C *vs.* 90 °C water conditions at similar time points including 2 min, 1 h, and 2 h; FRAP assay: (D) 27 °C water; (E) 90 °C water; (F) 27 °C *vs.* 90 °C conditions at the same time points; ABTS assay: (G) 27 °C water; (H) 90 °C water; (I) 27 °C *vs.* 90 °C water conditions at the same time points; ORAC assay: (J) 27 °C water; (K) 90 °C water; (L) 27 °C *vs.* 90 °C water conditions at the same time point. Data are represented in means ± SD ($n = 5$). The significant difference was determined using one-way ANOVA (*$p < 0.05$, **$p < 0.01$, ***$p < 0.001$) compared to the other group.

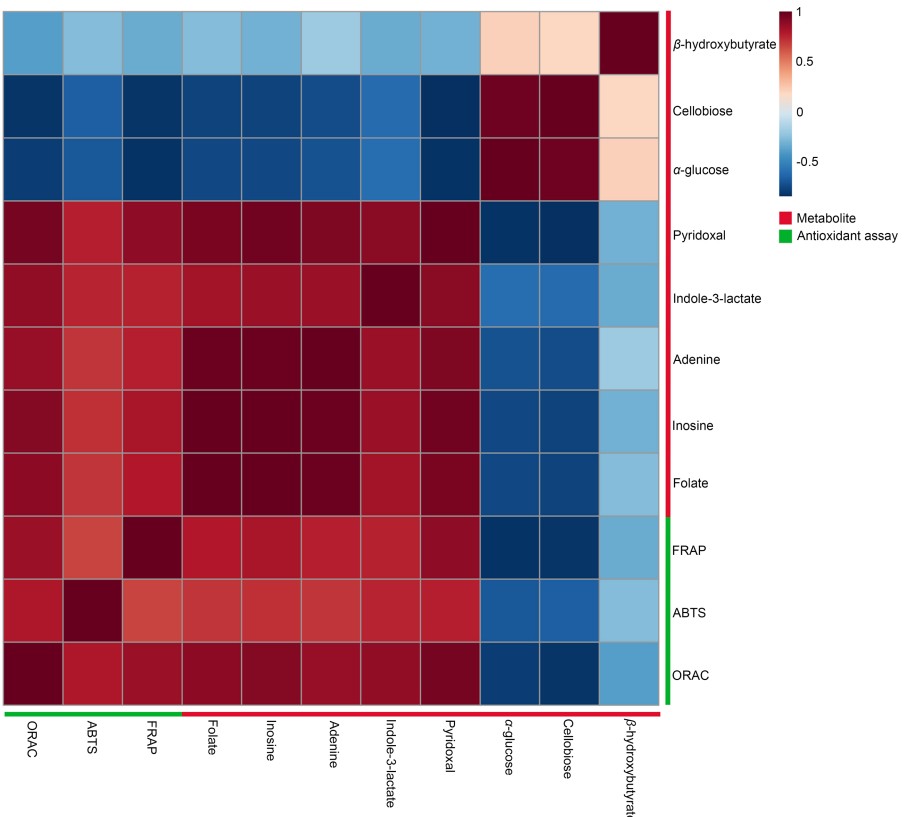

**Figure 5 Heatmaps of Pearson correlation between metabolites and antioxidant tests.** The heatmaps display Pearson correlation coefficients ($r$) between significant metabolites and antioxidant activities (FRAP, ABTS, ORAC). Red signifies positive correlations, while blue denotes negative correlations, with color intensity representing the strength of the relationships. Data were analyzed *via* MetaboAnalyst 6.0, with significance established at $p < 0.05$.

## DISCUSSION

The study investigated the recovery yield of aqueous-extracted dill leaves under different circumstances to see how temperature and extraction duration affected the quantity of dill extracts. Dill was subjected to aqueous extraction at both room temperature (27 °C) and hot temperature (90 °C) for various time points (2 min, 1 h, and 2 h) to assess if either variable influenced extraction efficiency. Our results indicated that an extraction yield was around 13–14%, with no variation across the different time or temperature conditions. This

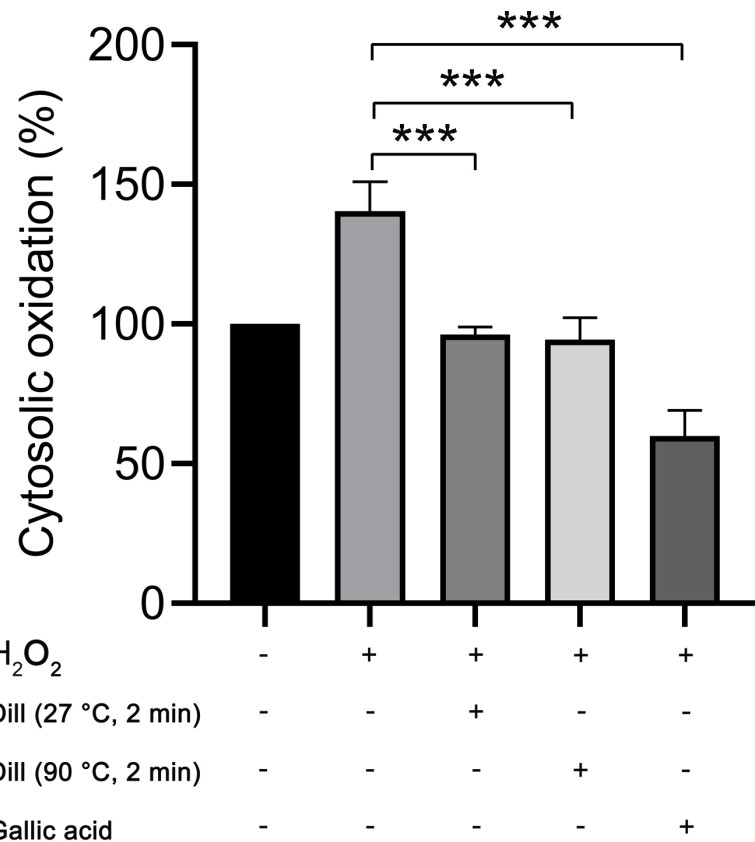

**Figure 6** **Percentage of cytosolic oxidation in HDFs after treatment with $H_2O_2$, extract of dill leaves, and gallic acid was measured using the CM-$H_2$ DCFDA technique, which detects an intracellular ROS.** Data are presented as means $\pm$ SD from three independent experiments. The significant difference was determined using one-way ANOVA (***$p < 0.05$) compared to control group.

finding suggests that neither higher temperature nor prolonged extraction time notably increases the yield of dill extracts.

The extractions results indicated an average percentage yield (g extract/g dried plant) of around 13–14% with no significant differences observed in the dry weight of dill leaf extracts. Additionally, temperature and time did not affect the quantity of dill leaf extracts obtained. A prior investigation employing the same maceration technique yielded lower average percentage yield (g extract/g dried plant) compared to our finding, specifically 0.06%, 0.04%, and 0.06% for water, ethanol, and acetone extractions, respectively (*Isbilir & Sagiroglu, 2011*).

Recent advances in metabolomics and chemometric approaches have enabled larger datasets of metabolites to be examined and analyzed (*Salem et al., 2020*); hence, our study utilized [1]H NMR-based metabolomics together with multivariate analysis to explore the metabolic profiles in aqueous-extracted dill leaves under various conditions. An unsupervised PCA indicated that dill leaf extracts under varying conditions can be differentiated by the first principal component (PC1). Consequently, we performed

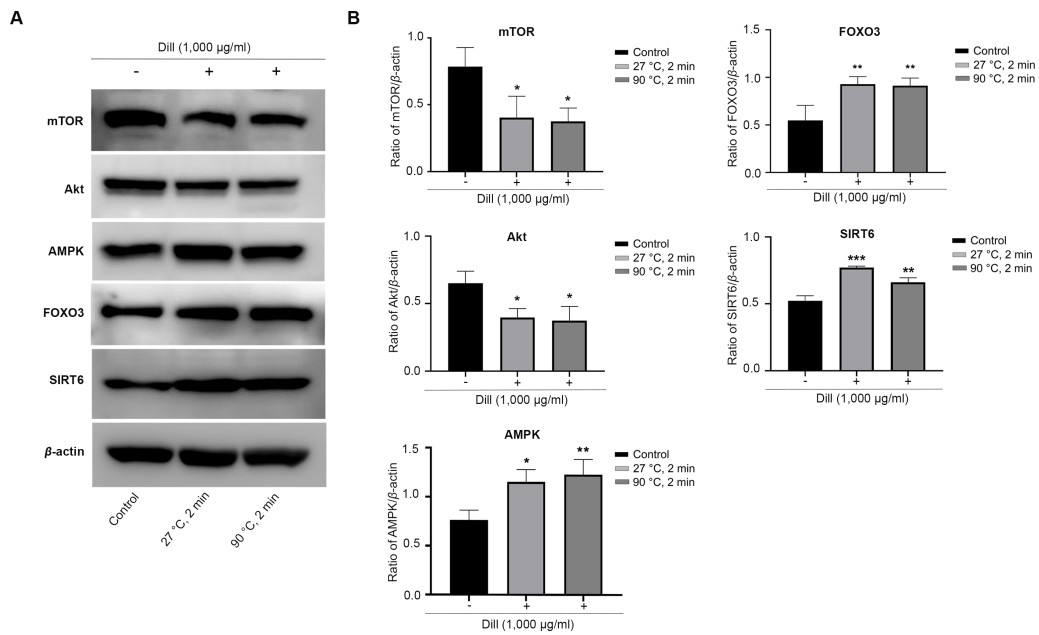

**Figure 7** **Effect of dill extracts of leaves on expression of longevity-related proteins in HDFs.** (A) Western blot analysis showed protein levels of mTOR, Akt, AMPK, FOXO3, and SIRT6 (B) The band signal intensity ratio of mTOR, Akt, AMPK, FOXO3, and SIRT6 protein per β-actin. Data are represented in means ± SD of three independent experiments. The significant difference was determined using one-way ANOVA (*$p < 0.05$, **$p < 0.01$, ***$p < 0.001$) compared to control group.

pairwise OPLS-DA to highlight the separation between the two conditions of dill leaf extracts. Next, we confirmed 41 identified dill metabolites using 2D $J$-resolved [1]H NMR analysis. Metabolites were classified into phytochemicals categories according to their roles in plant metabolisms, specifically as primary and secondary metabolites (*Rabizadeh et al., 2022*). Among the 39 primary metabolites of dill leaf extracts identified, 13 amino acids were found. Among those, seven essential amino acids (isoleucine, leucine, methionine, phenylalanine, threonine, tryptophan, and valine) were listed. In addition, two fatty acids, α-hydroxyisovalerate and isovalerate, were detected in aqueous extracts. Water-soluble vitamins and derivatives were found, including pantothenate (B5), pyridoxal (B6), pyridoxamine (B6), and folate (B9). The γ-aminobutyrate (GABA), which plays a major role in controlling anxiety, stress, and fear (*Möhler, 2012*), was also found. Three types of sugars, α-glucose, β-glucose, and sucrose, were detected. Interestingly, two secondary metabolites, indole-3-lactate and 3,7-dimethylurate, were identified. These secondary metabolites have been reported in the FooDB (FOOD00013) database (https://foodb.ca/foods/FOOD00013). The biological function of 3,7-dimethylurate remains unclear. On the other hand, indole-3-lactate contains anti-inflammatory properties (*Kim, Kim & Sung, 2022*), and it is commonly found as a major component in probiotic bacteria (*Zhou et al., 2022*).

The quantification of the absolute concentrations of dill-identified metabolites under varying conditions showed that temperature and extraction duration influence metabolic profile, affecting both the level and type of metabolites released. Interestingly, 90 °C

water extracts released a lower concentration of α- and β-glucose than those with 27 °C water conditions. These data show that dill leaves immersed in 90 °C water are beneficial for the consumer's health. Previous studies showed beneficial effects of dill on clinical and metabolic status in patients with type 2 diabetes (*Goodarzi et al., 2016*; *Haidari et al., 2020*). Our study identified 41 metabolites in the metabolic profiles of aqueous-extracted dill leaves, as determined by the $^{1}$H NMR metabolomic approach, which are implicated in various metabolic processes. Previously, they employed UHPLC-MS and GC-MS to investigate the metabolic profiles of dill leaf extracts in water through a targeted metabolomic approach (*Choe et al., 2023*). Those methods identified 10 polyphenol metabolites in dill that can disrupt the entry of SARS-CoV-2 into host cells. The types of metabolites identified differ from those in our study, which employed an untargeted metabolomics approach. Furthermore, untargeted metabolomic analysis using GC-MS identified 57 identified metabolites, of which 12 overlapped with our protocol. These included valine, leucine, succinate, fumarate, threonine, alanine, malate, phenylalanine, citrate, glucose, indole-3-lactate, and sucrose (*Castro-Alves et al., 2021*).

Antioxidant capacities within dill leaf extracts were tested. The results indicated that there were important differences in antioxidant activities between dill leaves extracted with water at 27 °C and those extracted with water at 90 °C in four different tests. Four antioxidant assays revealed that 90 °C water conditions had antioxidant activities greater than 27 °C water conditions. The antioxidant activities of dill leaf extracts were assessed using TPC and FRAP assays, which are based on single electron transfer (SET). The ORAC assay was employed to assess the antioxidant activity through hydrogen atom transfer (HAT). The ABTS assay was founded on the combination of SET and HAT mechanisms (*Prior, Wu & Schaich, 2005*). The results demonstrated that water-extracted dill leaf exhibited antioxidant activities, though these were at varying levels when compared to phenolic compounds that are more soluble in organic solvents (*Sultana, Anwar & Ashraf, 2009*).

Our study utilized one-way ANOVA and Pearson correlation to identify five metabolites that exhibit strongly positive correlations to antioxidant activities: pyridoxal, indole-3-lactate, adenine, inosine, and folate. Pyridoxal, a derivative of vitamin B6, serves as a component of coenzymes involved in various metabolic processes, including amino acid metabolism, hemoglobin synthesis, and neurotransmitter production (*Tambasco-Studart et al., 2005*). Pyridoxal is crucial for one-carbon metabolism and is involved in DNA methylation processes (*Mikkelsen et al., 2023*) associated with aging and longevity (*Salas-Pérez et al., 2019*). Indole-3-lactate has been previously reported as a component in dill. Indole-3-lactate functions as an anti-inflammatory agent for atopic dermatitis (AD) in skin-equivalent models (*Kim, Kim & Sung, 2022*) and serves as a mediator in the interaction between the host microbiota and neurons *via* bifidobacteria development (*Wong et al., 2020*). Furthermore, indole-3-lactate can enhance the growth of probiotics such as *Bifidobacterium* and *Faecalibacterium*. Their presence in the gut is considered a marker of a healthy microbiome (*Zhou et al., 2022*). Adenine serves as a substrate in the biosynthetic pathways of AMP, with the salvage pathway employing adenine phosphoribosyltransferase (Aprt) to facilitate the conversion of adenine to AMP. AMP levels rise in low-energy conditions, triggering AMPK activation linked to the longevity processes (*Stenesen et al.,*

*2013*). Inosine has strong immunomodulatory and neuroprotective properties, according to previous research (*Haskó, Sitkovsky & Szabó, 2004*). Additionally, inosine can enhance plant growth, especially root growth (*Tokuhisa et al., 2010*). A prior study indicated that low-dose folate supplementation exerts beneficial antioxidant effects, enhancing the lifespan of *Caenorhabditis elegans* by inhibiting the insulin/insulin growth factor 1 (IGF-1) and mechanistic target of rapamycin (mTOR) signaling pathways (*Rathor et al., 2015*).

Oxidative stress (OS) is a common phenomenon resulting from an imbalance between antioxidants and reactive oxygen species (ROS). Antioxidants are molecules or compounds that slow down, regulate, or stop oxidative reactions. They are found in very low concentrations in food and the body. Moreover, antioxidants play an important role in maintaining human health or promoting health, preventing, and treating diseases (*Shahidi & Zhong, 2015*). Therefore, we conducted a study on the longevity-inducing effect of dill leaf extracts on HDFs. Our findings demonstrated that preincubation of HDFs with dill leaf extracts followed by $H_2O_2$ treatment could considerably reduce the percentage of cytosolic oxidation. No significant differences were observed in the intracellular antioxidant activities of dill leaf extracts at varying temperatures in HDFs. Moreover, a previous study has also shown that dill extracts were not toxic to skin fibroblast cell lines but induced normal proliferation and cell growth (*Amin, 2022*). The expression of longevity-related proteins mTOR, Akt, AMPK, SIRT6, and FOXO3 were examined in HDFs treated with dill extracts in water at 27 °C and 90 °C for 2 min. A prior study indicated that oxidative stress activated the mammalian/mechanistic target of rapamycin (mTOR) through protein kinase B (Akt), highlighting the role of mTOR as a central regulator associated with proliferation, growth, and survival (*Papadopoli et al., 2019*). The AMPK, an important energy sensor, regulates cell cycle and autophagy, which is involved in longevity pathways (*Vellai, 2009*). Furthermore, the downstream effector of mTOR/Akt and AMPK proteins, FOXO3, is crucial in regulating various processes associated with longevity (*Wang, Zhou & Graves, 2014*). Moreover, SIRT6 reactivation might delay the process of aging in tissues (*Xu et al., 2015*). Triggered SIRT6 function commits to human longevity by promoting genome maintenance (*Simon et al., 2022*). Our results indicated that dill leaf extracts, when applied for 2 min at varying temperatures decreased the expression of mTOR and Akt while increasing the expression of AMPK, FOXO3, and SIRT6, which are crucial component of the longevity pathway.

As a result, dill shows antioxidant activities and also has biological functions that can promote health and therapeutic supplementation. A prior study examined dill powder encapsulated in capsules in clinical trials involving patients with type 2 diabetes. Dill powder supplementation resulted in a significant reduction in mean serum levels of insulin, homeostatic model assessment of insulin resistance, low-density lipoprotein cholesterol (LDL), total cholesterol, and malondialdehyde, while simultaneously increasing high-density lipoprotein (HDL) and total antioxidant capacity. Therefore, dill intake might be useful in the treatment of diabetes complications (*Haidari et al., 2020*). Moreover, dill contains branched-chain amino acids (BCAAs), which are a complex of three essential amino acids: leucine, isoleucine, and valine. BCAAs are an essential component that accounts for one-third of muscle proteins, and they must be obtained from dietary sources

that cannot be synthesized *de novo*. BCAAs can reduce the feeling of delayed muscle pain, which follows because of muscle damage (*Tambalis & Arnaoutis, 2022*). Therefore, dill leaves might be recommended as a dietary supplement for enhanced well-being.

## CONCLUSION

The metabolomics study revealed that aqueous-extracted dill leaves with various time points and temperature conditions released different concentrations of metabolites. Results indicated that temperature-related and extraction duration-related factors affected the level of metabolites. Dill leaves also contain high levels of nutraceuticals and strong antioxidant properties at 2 min in 90 °C water. Dill leaves possess antioxidant properties and biological activities that protect against oxidative damage. They can also extend the lifespan of human dermal fibroblasts after exposure to oxidative stress. All our findings suggest that dill could promote health and might develop into therapeutic supplementation in the future. Therefore, dill intake may prove useful for people who often consume dill when cooking or drinking tea.

## ACKNOWLEDGEMENTS

We would like to acknowledge Professor Ross H Andrews for editing this manuscript.

### Funding

This work was supported by a grant from Faculty of Medicine, Khon Kaen University (grant. no. IN66080) and a Postgraduate Study Support Grant of Faculty of Medicine, Khon Kaen University to JT and the NSRF under the Fundamental Fund of Khon Kaen University to NN. The funders had no role in study design, data collection and analysis, decision to publish, or preparation of the manuscript.

### Grant Disclosures

The following grant information was disclosed by the authors:
Faculty of Medicine, Khon Kaen University: IN66080.
Postgraduate Study Support Grant of Faculty of Medicine, Khon Kaen University.
Fundamental Fund of Khon Kaen University.

### Competing Interests

The authors declare there are no competing interests.

### Author Contributions

- Jirattiporn Thanuma conceived and designed the experiments, performed the experiments, analyzed the data, prepared figures and/or tables, authored or reviewed drafts of the article, and approved the final draft.
- Jutarop Phetcharaburanin analyzed the data, authored or reviewed drafts of the article, and approved the final draft.

- Hasaya Dokduang performed the experiments, authored or reviewed drafts of the article, and approved the final draft.
- Watcharin Loilome conceived and designed the experiments, authored or reviewed drafts of the article, and approved the final draft.
- Poramate Klanrit conceived and designed the experiments, authored or reviewed drafts of the article, and approved the final draft.
- Arporn Wangwiwatsin conceived and designed the experiments, authored or reviewed drafts of the article, and approved the final draft.
- Nisana Namwat conceived and designed the experiments, prepared figures and/or tables, authored or reviewed drafts of the article, and approved the final draft.

### Data Availability

The raw data are available in the Supplemental Files.

### Supplemental Information

Supplemental information for this article can be found online at http://dx.doi.org/10.7717/peerj.19567#supplemental-information.

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
