# Peer review of "Metabolomic analysis of bioactive compounds in dill (Anethum graveolens L.) extracts"

_PeerJ, doi:10.7717/peerj.19567_

## Round 0.1 · original submission · Major Revisions

The manuscript was reviewed by three independent experts in the field. All the reviewers found the work interesting but raised several issues which should be addressed for further consideration. The reviewers provide detailed comments in their reviews and point out the areas where the manuscript needs to be improved.

**Language Note:** The review process has identified that the English language must be improved. PeerJ can provide language editing services - please contact us at [email protected] for pricing (be sure to provide your manuscript number and title). Alternatively, you should make your own arrangements to improve the language quality and provide details in your response letter. – PeerJ Staff

·

Basic reporting

Please consider declaring ROS as reactive oxygen species in the abstract to increase readability.

Experimental design

1. Please explain the rationale for selecting 27C and 90C as extraction temperatures.

2. Please include a paragraph to introduce the current available methods to quantify antioxidant capacities and why the four methods (TPC, FRAP, ABTS, ORAC) were selected in this study.

Validity of the findings

-

Additional comments

The authors are advised to consider putting figures 2, 5, and 6 into the supplemental information. A more concise visualization layout would highlight the key messages conveyed in the study.

Reviewer 2 ·

Basic reporting

The manuscript presents an investigation of the bioactive metabolites present in the extract of dill leaves. However, the manuscript can be improved for clarity and magnify the figures according to the style of the journal.

Experimental design

The study design and experimental procedures are unclear and need references.

Validity of the findings

Results are valid, but the tables or figures summarizing key findings would aid in interpreting the data and enhance the reader's understanding. Figures need magnification. Authors should keep in mind the style of the journal

Additional comments

Authors should update the references of the manuscript, and English language is poor

Annotated reviews are not available for download in order to protect the identity of reviewers who chose to remain anonymous.

Reviewer 3 ·

Basic reporting

- Language must be improved where several sentences are not readable and complicated.
- The paragraphs in the article must be properly formatted.
- The references appear to be generally well-formatted, but they need some adjustments to ensure consistency in style. Here are some notes:
- Year placement: Sometimes, the year appears right after the names, and in other cases, after the title. It should consistently follow the authors’ names.
- Title formatting: Some titles are written in all capital letters (Capitalized), while others use lowercase. It is best to follow a consistent style, such as capitalizing only the first word and proper nouns.
- Journal and periodical names: Some references use abbreviated journal names, while others use full names. Standardize this throughout.
- Formatting of volume and page numbers: Some references have extra spaces or inconsistent symbols when listing page numbers and volumes.
- DOI formatting: Some references present DOI as a direct link, while others provide only the number. It is best to use a consistent format.

Experimental design

- Why did you choose two variables in this study?
- What is the rationale behind selecting these specific time intervals: 2 min, 1 h, and 2 h for the time variable in this study?
- Please add the collection site of dill.

Validity of the findings

- The accuracy of the image Figures in the article should be improved.

---

## Round 0.2 · accepted · Accept

I thank the authors for revising the manuscript satisfactorily. The current version is suitable for the publication.

·

Basic reporting

-

Experimental design

-

Validity of the findings

-

Additional comments

-

Reviewer 2 ·

Basic reporting

All revisions were made. The Manuscript is acceted

Experimental design

It is ok

Validity of the findings

The manuscript is accepted

Additional comments

The manuscript is accepted